# Structured Observations of Child Behaviors during a Mastery-Motivational Climate Motor Skill Intervention: An Exploratory Study

**DOI:** 10.3390/ijerph192315484

**Published:** 2022-11-22

**Authors:** Kara K. Palmer, Emily R. Cox, Katherine Q. Scott-Andrews, Leah E. Robinson

**Affiliations:** 1Child Movement, Activity, and Developmental Health Laboratory, School of Kinesiology, University of Michigan, Ann Arbor, MI 48109, USA; 2Health Studies Department, College of Arts and Sciences, American University, Washington, DC 20016, USA

**Keywords:** engagement, program, behavior, exploratory factor analysis

## Abstract

This exploratory study aimed to quantify children’s engagement behaviors during a mastery-motivational climate intervention. We also completed an exploratory factor analysis to elucidate if child engagement changed across intervention sessions. Method: 35 children (17 boys; 18 girls) completed a 10-week mastery-motivational climate motor skill intervention. Engagement was operationalized as the time children were appropriately involved in the intervention and was assessed using momentary time sampling during the motor skill practice portion of the intervention. Results: Overall, children were engaged 36% of the motor skills practice time (37% for boys; 36% for girls). Children who initially had below-average skills engaged for 36% (36% for boys; 35% for girls) of the motor skills practice time, and children who were average or above-average at the start of the intervention engaged in skill practice for 39% (39% for boys; 36% for girls). Differences in engagement in skill type (e.g., locomotor vs. ball skills) and trends over time were observed. Conclusion: These findings support that children engage in mastery-motivation climates, but the amount of participation may be influenced by individual factors of sex and initial skill level.

## 1. Introduction

Fundamental motor skills are gross motor skills that provide a foundation for future, more complex or sport-specific movements [1]. These skills are frequently divided into locomotor skills, skills that propel the body in space (e.g., run, skip, or jump), ball skills, skills that propel or manipulate objects in space (e.g., throw, kick, catch), or balance skills, skills that maintain upright posture [2]. These skills are essential for children’s current and future health and development [3,4]. Unfortunately, children do not appear to learn these skills without specific interventions or programming [3,5]. Instead, these skills must be “taught, practiced, and reinforced” [6].

One intervention approach shown to improve motor skills is using mastery-motivational climates for motor skill instruction [7,8]. These climates are grounded in achievement goal theory and apply the six TARGET (Task, Authority, Recognition, Grouping, Evaluation, and Time [9]) structures to the intervention setting. A child’s autonomy to self-navigate the intervention climate is central to these structures. The child can decide what level of difficulty to practice (*task*), their peers or peer groups to play with (*grouping*), and how long they will spend at any given station (*time*). The instructor acts as a facilitator, not a dictator of the learning environment, and shares the classroom leadership with the children (*authority*). Instructors engage with each child individually to gauge a child’s individual learning (*recognition*) and to assess a child’s growth and development based on self-referenced standards (*evaluation*). These climates can appear more chaotic and less controlled than other types of learning environments due to the increased autonomy granted to the children. However, perhaps counterintuitively, these approaches to skill instruction are remarkably effective for teaching motor skills to young children even with the lack of “control” from teachers. Children demonstrate immediate and sustained skill improvements after engaging in interventions that adopt this theoretical approach to instruction (see the following for a full review; [7,8]).

Interestingly, while we know these interventions improve skills, there is a dearth of knowledge on child behaviors during these interventions themselves. We know only two studies examined children’s behaviors during a mastery-motivational climate. The first study quantified children’s engagement as practice attempts and reported the number of practice attempts children completed in high-autonomy and low-autonomy interventions [10]. Results found no statistically significant difference in the number of skill attempts based on climate, but children in the low-autonomy group completed more attempts than the high-autonomy group [10]. Learning as displayed through motor skill change across both interventions was the same, but children in the high-autonomy group had gains in perceived motor competence. In contrast, children in the low-autonomy group did not see changes in their perceived motor competence [10].

The second study applied a strength-based approach or “appreciative inquiry” to illuminate the pathway 4-year-old children follow from novice to skilled learners in a 9-month mastery-motivational climate intervention [11]. This research design focuses on discovering how individuals or lesson climates help participants in gaining success [12,13]. The qualitative data collected in this strength-based approach included teacher interviews with the lead interventionist and photographs of the children engaged during each session. Results found that children act in three distinct stages as their skills progressed across the intervention—*captivation and exploration, cooperation and consolidation,* and *dedication and collaboration*. During the first stage, *captivation and exploration*, the children were enthralled by the freedom to move. They enjoyed the novelty of new “play” equipment while learning the rules and safety of the environment. The teacher noted that children were off-task most of the time during this stage, with few moments of on-task behaviors. During the second stage, *cooperation and consolidation*, children engaged in tasks the way they were intended to and understood the purpose of each station while also playing alongside other children. The teacher was more involved in the lesson itself instead of behavior management and redirecting children to more on-task behaviors. During the third stage, *dedication and collaboration*, the children sought out and engaged the teacher to improve their movement and experienced fully organized team play. During this stage, the children’s behaviors allowed the teacher to spend more class time working on individual feedback and correction. This qualitative work supports that children’s behaviors change across an intervention; however, these conclusions were derived from teacher reports of child behaviors and not structured observations of child behaviors themselves.

Cumulatively, this work shows that child behaviors are important and may change across an intervention and demonstrates the need for research to better understand child behaviors in interventions. Additional work is needed that directly observes child behaviors across an intervention and could provide added knowledge into how children engage in mastery-motivational climates. Therefore, the purpose of this exploratory study was to directly observe and quantify children’s engagement during a mastery-motivational climate intervention. Further, since it appears that children’s behavior may change across the intervention itself [11], we completed an exploratory factor analysis to determine if there are changes in child engagement across the intervention.

Additionally, this research supports the need to account for how children’s individual factors may or may not influence their behaviors during an intervention. The importance of individual factors is evident from both a theoretical and practical perspective. The constraints-based model of movement posits that movement emerges from the interaction of environmental, task, and individual constraints or factors [14]. The creation of mastery-motivational climate interventions strategically manipulates the environmental and task constraints to promote certain functional individual constraints such as motivation. However, children still enter the environment with their own individual factors or constraints that are less malleable or structural that may influence their movement. Two important factors to consider may be sex and initial skill level. There are well-documented sex differences whereby boys outperform girls in most motor skills, especially ball skills [15,16]. While there is evidence that both boys and girls demonstrate equivalent gains in both locomotor [17] and ball skills [17,18] across an intervention, limited evidence exists on how boys and girls engage within the intervention. Secondly, some work supports that children’s initial skill level at the start of the intervention influences their engagement, whereby more highly skilled children complete fewer skill practice attempts than less skilled children [10]. Therefore, this study examined if children’s engagement overall or across the intervention differed based on individual constraints of sex and/or initial skill level. Understanding child engagement may lead to important points of future teacher training or individual intervention strategies to maximize intervention effects.

## 2. Materials and Methods

### 2.1. Participants

Children were recruited from a single Head Start Center in a large urban Midwestern city in the United States. All children from four randomly selected classrooms who were 3.5 years or older and did not have a physical or cognitive disability document in their school records were invited to participate in the study. Thirty-five children (18 girls; *M*_age_ = 4.5 ± 0.27 yrs; 54.3% African American, 14.3% White, 5.7% Latinx, and 25.7% other) provided parental consent and verbal assent and were enrolled as participants.

### 2.2. Children’s Health Activity Motor Program (CHAMP)

CHAMP is an evidence-based mastery-motivational climate intervention that enhances motor skills [18,19,20], perceived competence [6], physical activity [21,22], and self-regulation [20] in young children. CHAMP is grounded in achievement goal theory [9,23,24] and adheres to Epstein’s TARGET structures. CHAMP intervention grants children autonomy to self-navigate through and to dictate their own engagement in the intervention’s learning objectives, in this case, motor skill instruction. While children are always encouraged to be engaged in the intervention learning objectives/skill practice, there are no restrictions placed on children in regard to their peer groups (*grouping*), time spent at each motor skill station (*time*), nor what level of difficulty they practice a task (*task*). Instructors facilitate the session (*authority*) and offer children individualized and private praise and encouragement (*recognition* and *evaluation*).

Children replaced their regularly scheduled standard practice (outdoor free play) session with CHAMP 3 days a week for 10 weeks (45 min/day × 30 sessions = 1350 min). Each session was 45 min in duration and included: (a) 2–3 min of introduction and warm-up; (b) 3–5 min of motor skill station description; (c) 27–33 min of motor skill stations practice and independent instruction; (d) 3–5 min closing. Each session, children received access to 3–4 stations (minimum of one locomotor and one ball skill), each with 3–4 differentiated levels of difficulty. Two expert instructors delivered all intervention sessions (one lead and one assistant, each with graduate-level education in pediatric motor development) and one support staff. A support staff member completed a daily fidelity check to ensure that the intervention was implemented correctly and adhered to the TARGET structures. The intervention was implemented with 100% fidelity to TARGET structures and 98% fidelity for instruction.

### 2.3. Structured Observation of Engagement

Engagement was operationalized as the time children were actively and appropriately involved in the intervention. A child was considered “engaged” when they were practicing motor skills, gathering/preparing equipment to use, actively watching peers complete the skill, intentionally transitioning between skill stations, or receiving skill instruction. Children were not considered “engaged” when they were in off-task play, socializing, disengaged or emotionally unregulated, or standing idly. Engagement was examined during the intervention’s motor skill stations practice and independent instruction portion of CHAMP. Engagement was not assessed during the introduction and closing activities. The primary outcome was the percentage of time children were engaged and was quantified overall and for both locomotor and ball skills separately.

All intervention sessions were video recorded, and engagement was assessed via structured observation of the videos using a momentary time sampling technique with a 10-s observation, 5-s record coding interval [25,26,27]. One-minute observations (four, 15-s coding intervals) alternated between randomly selected boys and girls. Between 11 and 12 children were recorded daily (i.e., approximately four observations per child across a single intervention session), and each child was recorded at least once a week. If a child was absent from the session they were selected to be recorded, researchers would randomly select a different session that week to observe the child. Researchers listened to a timer available through the mobile phone app Data Timer^®^ that gave auditory signals to observe and record. Two coders underwent a 3 h training before coding the data and established an inter-rater agreement of 90% before data coding started. Once coding was underway, coders watched all videos, scored engagement (25% overlap), and demonstrated a >99% agreement on overlapping coded sessions.

### 2.4. Procedures

All study procedures were approved by the Institutional Review Board (HUM00135602). Children provided both parental written consent and child assent before being enrolled in the study. All children completed anthropometric and motor skill assessments (i.e., Test of Gross Motor Development-3rd Edition (TGMD-3)) before (pretest) and after (posttest) the 10-week CHAMP intervention. Structured observations of children’s engagement in the intervention were recorded for each of the 30 intervention sessions, and each child was observed weekly.

### 2.5. Analyses

Due to the exploratory purpose of this study, statistical analyses were not conducted to confirm or reject pre-determined hypotheses. Instead, descriptive data from structured observations were recorded and reported. Data were plotted into graphics to examine engagement across the intervention, and an exploratory content analysis was completed on the graphics to potentially identify trends or phases in children’s engagement behaviors across time [28]. Engagement was reported, and graphics were created for all children, children divided by sex (boys or girls), and children divided by skill level at pretest (below-average or average and above-average) for total, locomotor, and ball skills. Skill level at pretest was determined using total pretest TGMD-3 index score, and children were categorized into two groups: below-average or average and above-average. Children with an index score of 89 or below were considered below-average and children with an index score of 90 or above were considered average and above-average [29]. After creating the graphics, plots were separately inspected by two authors (EC and KP), and identified trends were discussed. Any discrepancies in interpretations were further discussed until a consensus was reached.

## 3. Results

### 3.1. Sample

Children were subdivided on initial skill level, with 21 children categorized as below-average (11 girls, 10 boys) and 14 children categorized as average or above-average (7 girls, 7 boys).

### 3.2. Overall Intervention Engagement

A total of 3403 observations were completed on all 35 children (18 girls, 17 boys) across 30 intervention sessions. On average, 215.23 observations (53.81 min) were completed on each child. Each CHAMP session included an average of 27.23 ± 3.96 min of motor skill stations practice and independent instruction.

Overall, children were engaged for 36% (16% locomotor; 20% ball skills) of the motor skill stations practice and independent instruction. Girls engaged for 36% (19% locomotor; 17% ball skills) of the time, and boys engaged for 37% (13% locomotor; 24% ball skills) of the time. Children who had below-average skills at the start of the intervention engaged for 36% (18% locomotor; 17% ball skills) of the motor skill stations practice and independent instruction. Children with average and above-average skill level at the start of the intervention engaged for 39% (13% locomotor; 25% ball skills) of the time. When divided by both sex and initial skill level, boys who had below-average skills at the start of the intervention engaged for 36% (17% locomotor; 19% ball skills) of the motor skill stations practice and independent instruction, and boys with average and above-average at the start of the intervention engaged for 39% (9% locomotor; 30% ball skills) of the time. For girls, those with below-average skills at the start of the intervention engaged for 35% (20% locomotor; 16% ball skills) of the activity session, and girls with average and above-average at the start of the intervention engaged in positive intervention behaviors for 36% (17% locomotor; 19% ball skills) of the time.

### 3.3. Exploratory Content Analyses Examining Intervention Engagement over Time

Overall, there was a slight decrease in children’s engagement across the intervention (Figure 1A). For locomotor skills, children’s engagement decreased over time, with a slight increase across the last four sessions (Figure 1B). For ball skills, children’s engagement appears to be an inverted-U shape, where children engaged more with ball skills in the middle of the intervention than at the beginning or the end of the intervention (Figure 1C). See Figure 1 for graphs examining engagement for all children.

### 3.4. Exploratory Content Analyses of Engagement by Sex

When examining the data by sex, there was a similar pattern for boys, with overall skills decreasing over time and an inverted-U shape for engagement in ball skills (see Figure 2A,C). However, boys had low engagement in locomotor skills with a few peak days across the intervention (see Figure 2B). Girls had a similar decrease in skills overall (see Figure 3A); but they were more consistently engaged in locomotor skills (see Figure 3B) and appear to have two peaks in engagement in ball skills (peak days 6 and 21; see Figure 3C).

### 3.5. Exploratory Content Analyses of Engagement by Initial Skill Level

When examining the data by initial skill level, children with below-average skills decreased engagement over time, whereas children with average or above-average skills appear to have two inverted-U shape peaks in engagement (peak days 3 and 21; see Figure 1D). Further, children seem to have similar amounts of engagement regardless of initial skill level at the beginning and end of the intervention, but children with average or above-average skills had higher engagement on most days in the middle of the intervention (days 14–25; see Figure 1D). When considering just locomotor skills, there was an overall decrease in engagement in locomotor skills across the intervention (see Figure 1E). Children with average and above-average skills were consistently less engaged in these skills than children with below-average skills with a few exceptions (days 19 and 20; see Figure 1E). There was an overall inverted-U shape to children’s engagement when considering just ball skills, but children with average and above-average skills engaged in more ball skills than children with below-average skills (see Figure 1F). Further, the pattern of engagement over time appears to be more consistent for children with below-average skills but more variable, with higher peaks and valleys for children with average and above-average skills (see Figure 1D–F).

### 3.6. Exploratory Content Analyses on Sex by Initial Skill Level

When examining sex by initial skill level interactions, boys with below-average skills had a decrease in engagement over time (see Figure 2D). In contrast, boys with average and above-average skills had a more inverted-U shape to their engagement (see Figure 2D). There was a crossover so that boys with below-average skills appeared to engage more overall at the beginning of the intervention, and boys with average and above-average skills engaged more in the middle of the intervention (see Figure 2D). At the end of the intervention, all boys were equally engaged regardless of initial skill level (see Figure 2D). Boys with below-average skills engaged more overall for locomotor skills, but this engagement is demonstrated as a few peaks across the intervention (see Figure 2E). There were two inverted U’s for ball skills, but the peaks appear to be different based on initial skill level. Boys with below-average skills peaked at day 13 and day 27, whereas boys with average and above-average skills peaked at day 14 and day 25 (see Figure 2F). Boys with average and above-average skills consistently engaged in more ball skills compared with boys with below-average skills. See Figure 2 for full graphs of boys’ engagement.

For girls with below-average skills, there appears to be a slight inverted-U shape in their engagement across the intervention (see Figure 3D). Girls with average and above-average skills have two peaks across the intervention (day 3 and 19; see Figure 3D). In general, girls with average and above-average skills engage more than girls with below-average skills across the intervention for all skills (see Figure 3D). Girls with below-average skills engaged more with locomotor skills compared with girls with average and above-average skills with a few exceptions (see Figure 3E). For ball skills, girls with below-average skills had a single inverted-U shape in their engagement with a peak around day 21 (see Figure 3F). Girls with average and above-average skills had two inverted-U shapes with peaks at days 3 and 18 (see Figure 3F). See Figure 3 for full graphs of girls’ engagement.

## 4. Discussion

Mastery-motivational climates are an effective intervention approach to improve motor skills in young children [7,8], but less is known about how children engage within the intervention itself. This study directly observed and quantified children’s engagement during a mastery-motivational climate intervention. We also examined if children’s individual factors of sex and/or initial skill level influenced their engagement behaviors. Lastly, we completed an exploratory factor analysis to determine if there are changes in child engagement across the intervention itself based on previous research that reported three distinct learning phases across an intervention [11].

Results found that, on average, children were only actively engaged for 36% of the motor skill stations practice portion of the intervention. These results show that children are participating in skill practice for less than half of the allotted skill practice time and instead are using the high-autonomy climate to participate in a wide variety of behaviors. There is limited research to compare with our reported percentage of time engaged in the intervention, specifically within preschool mastery-motivational climates. In the physical education literature, students engage in active learning responses for 57% of the content time of a physical education lesson, of which 26% were motor responses [30]. Research examining children’s engagement in physical activity during physical education found that, on average, children in elementary school engage in moderate-to-vigorous physical activity for a similar percentage of class time (33.8% ± 13.6%, range = 8.9%–50.1%) when physical activity is quantified using direct observation [31]. The similarities in the percentage of time engaged in a particular behavior (e.g., engagement in positive motor skill behaviors, motor response, or physical activity), support that children use the less structured environment of physical education or mastery-motivational climates to engage in a wide variety of behaviors and activity levels.

Further, we found this level of engagement in CHAMP was comparable for both boys and girls overall (37% vs. 36%; respectively) as well as for children with average or above-average skills and children with below-average skills at the start of the invention (39% vs. 36%; respectively). The similar participation for both these groups aligns with previous mastery intervention research that supports that children complete the same amount of appropriate skill attempts in a ball skill mastery motivational climate intervention regardless of individual factors [10]. While engagement and learning are undoubtedly different outcomes, the similar levels of engagement between boys and girls parallels findings regarding improvements in skills across mastery-motivational climates regardless of sex [17,18]. Because girls often exhibit poorer motor skills than boys [15,18], the knowledge that girls will equally engage in and learn from interventions implemented using mastery-motivational climates is meaningful. When examining how boys and girls engaged in different types of skills, these results found the girls engaged fairly equally in both locomotor (19%) and ball skills (17%), whereas boys engaged less in locomotor skills (13%) compared to ball skills (24%). These results are interesting and show that even when afforded equitable opportunities to engage in different skills, boys will engage in more ball skills than locomotor skills and more ball skills than girls in the same intervention. Evidence does not support that these differences translate into learning differences [16,17] but could indicate established social-cultural biases [32].

Somewhat unexpectedly, we found differences when children’s engagement was examined using sex by initial skill level interaction. Our results showed that girls’ initial skill level was less likely to influence which type of skills they interacted with, but for boys, boys with higher initial skill levels engaged in more ball skills (30%) compared with locomotor skills (9%). These results could be interpreted using Newell’s constraint model and demonstrate how multiple individual factors, sex, and initial skill level, interact to influence skill behaviors [14]. These results also support that interventionists should consider the individual factors and the interaction among individual factors of children for the intervention climate.

We also examined how children’s engagement behaviors changed across the intervention. These results found an overall decrease in skill engagement with several inverted-U patterns for engagement in ball skills. Children’s engagement patterns appeared to differ by sex, initial skill level, and sex by initial skill level interactions. Previous literature supports that children’s behaviors change across an intervention in three phases- *captivation and exploration*, *cooperation and consolidation*, and *dedication and collaboration* [11]. These phases were derived from descriptions of child behaviors reported by the instructor and not directly from child behaviors themselves. While our findings did not show three distinct phases per se, these results are not contradictory to this earlier work and should be considered supplemental in that both teacher perspectives of children’s behaviors and direct quantifications of children’s behaviors provide unique and novel insight into what is happening during mastery-motivational interventions. The inverted-U shapes reported for some skills and groups may align with the three phases of *captivation and exploration* (increase in engagement behaviors), *cooperation and consolidation* (steady engagement behaviors), and a decrease in behaviors (*dedication and collaboration*). Cumulatively, data from both these studies support that children’s behaviors change across an intervention. The current results further show that engagement patterns appear to differ based on individual factors of sex and initial skill level.

It is important to note that the purpose of this study was to quantify children’s engagement, so we did not examine how engagement related to changes in outcomes. An established body of physical education literature supports that it is not the amount of skill practice but rather an engagement in motor-appropriate behaviors and not motor-inappropriate behaviors that are linked with changes in skills [33,34,35]. For this study, we were less interested in quantifying the type of skill practice (e.g., motor-appropriate vs. motor-inappropriate), so we did not relate engagement in CHAMP to changes in skills after the intervention. We were interested in examining children’s overall engagement behaviors to shine light into the proverbial “black box” of motor skill interventions and to better understand children’s behaviors during mastery motivational climates. These interventions have been established as an effective theoretical approach to designing and implementing motor skill interventions, but relatively little is known about children’s behaviors in these interventions. Understanding children’s behaviors during a mastery-motivational climate is fundamental as these climates are hallmarked by giving children autonomy to self-navigate the intervention sessions. Therefore, while this paper found differences in children’s engagement based on individual factors of sex and initial skill level, it remains unknown how this engagement is related to skill outcomes after the intervention.

This study included several strengths. First, we used structured observations of child behaviors from video-recorded data. This systematic approach to recording child behaviors ensures consistent sampling of data across the intervention and has been used in various assessments of student behaviors such as the System for Observing Fitness Instruction Time [26,27] or Children’s Activity Rating Scale [36,37]. Further, using video data ensured that researchers did not miss observations and could replay data if needed. Using video data also ensured that almost all children were observed weekly as sessions could be re-coded if a child was absent on the day they were originally assigned to be scored. This study also includes several limitations. First, we quantified engagement as the time children were actively involved in the intervention. A child was considered “engaged” when practicing the skill, gathering/preparing equipment to use, actively watching peers complete the skill, intentionally transitioning between skill stations, or receiving skill instruction. While this definition broadly aligns with previous literature [10], this definition and quantification is limited. Only children’s overall engagement was examined. As stated previously, the literature supports it is the type of practice (e.g., motor-appropriate vs. motor inappropriate) that is predictive of outcomes, so it may not be possible to link engagement as measured in this study with intervention outcomes. Nonetheless, this operational definition of engagement was adopted as it aligned with the study’s purpose to examine and quantify children’s behaviors within an intervention setting. This study was also limited in that children were only observed once a week and the final percentages presented represent average engagement overall. Lastly, no formal hypothesis testing was conducted, making causal claims or inferences regarding differences in group engagement is impossible. This decision was made in alignment with the exploratory purpose of this study, but it is important to note this limitation, especially when considering how data should be interpreted.

## 5. Conclusions

This study quantified children’s engagement in a mastery-motivational climate intervention using direct observation. Secondarily, this study explored if individual factors of sex and/or initial skill level impacted children’s engagement in the high-autonomy environment created during the mastery-motivational climate intervention. Results show that children are engaged in appropriate behaviors for approximately 36% of the “motor skill stations practice and independent instruction” part of the intervention. To the best of our knowledge, this research is the first to directly quantify children’s engagement during a mastery-motivational climate in a motor skill intervention for preschoolers. Results are similar to other research examining older children’s engagement slightly in physical education or physical activity during physical education classes. Further, this research found that children’s engagement does appear to differ according to sex by initial skill level interaction, particularly for boys. This research addresses a gap in knowledge by directly quantifying young children’s behaviors during a high-autonomy motor skill intervention and adds to our understanding of how these climates support and enhance children’s motor skills. In other words, this research shines a light on the proverbial “black box” of motor skills interventions, where we have information on the outcomes but not the processes. Future research is needed to determine how engagement might or might not relate to both immediate and sustained outcomes particularly based on children’s individual factors or sex or initial skill levels and analysis of quality of engagement.

## Figures and Tables

**Figure 1 ijerph-19-15484-f001:**
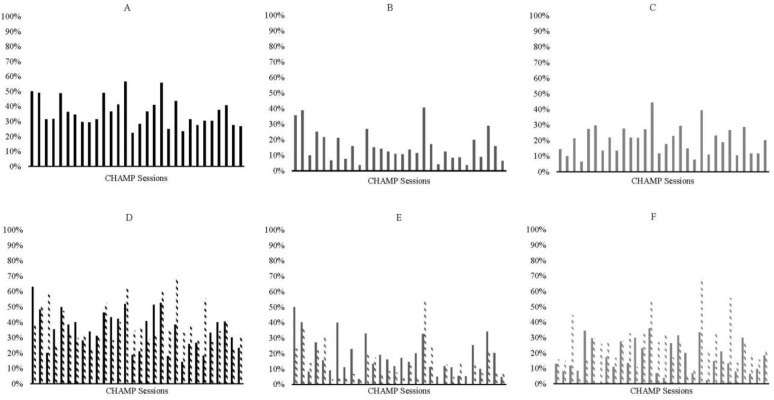
Graphs representing all children’s engagement across the 30-intervention sessions. In all plots, percentage of engagement is plotted on the y-axis and session number if on the x-axis. Plots include (**A**) all skills, (**B**) locomotor skills, (**C**) ball skills, (**D**) all skills for children with below average initial skills and average and above average initial skills, (**E**) locomotor skills for children with below average initial skills and average and above average initial skills, and (**F**) ball skills for children with below average initial skills and average and above average initial skills. For graphs (**D**–**F)**, children with below average initial skills are plotted in solid bars and children in average & above average initial are plotted in dashed bars.

**Figure 2 ijerph-19-15484-f002:**
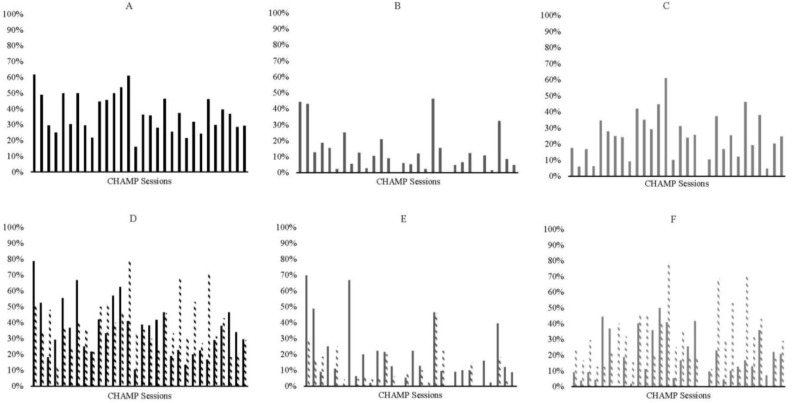
Graphs representing all boys’ engagement across the 30-intervention sessions. In all plots, percentage of engagement is plotted on the y-axis and session number if on the x-axis. Plots include (**A**) all skills, (**B**) locomotor skills, (**C**) ball skills, (**D**) all skills for children with below average initial skills and average and above average initial skills, (**E**) locomotor skills for children with below average initial skills and average and above average initial skills, and (**F**) ball skills for children with below average initial skills and average and above average initial skills. For graphs (**D**–**F)**, children with below average initial skills are plotted in solid bars and children in average & above average initial are plotted in dashed bars.

**Figure 3 ijerph-19-15484-f003:**
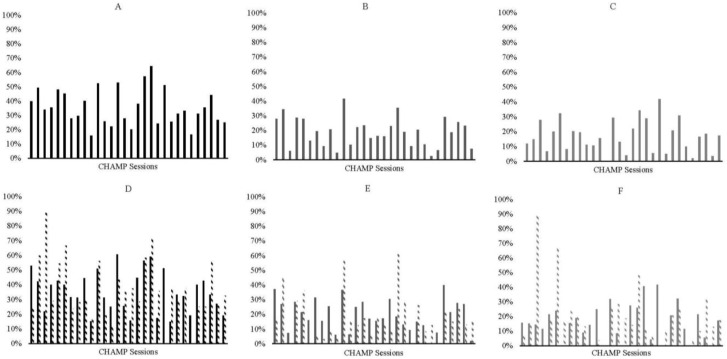
Graphs representing all girls’ engagement across the 30-intervention sessions. In all plots, percentage of engagement is plotted on the y-axis and session number if on the x-axis. Plots include (**A**) all skills, (**B**) locomotor skills, (**C**) ball skills, (**D**) all skills for children with below average initial skills and average and above average initial skills, (**E**) locomotor skills for children with below average initial skills and average and above average initial skills, and (**F**) ball skills for children with below average initial skills and average and above average initial skills. For graphs (**D**–**F)**, children with below average initial skills are plotted in solid bars and children in average & above average initial are plotted in dashed bars.

## Data Availability

The data presented in this study are available on request from the corresponding author. The data are not publicly available.

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
