# Peer review of "Structured Observations of Child Behaviors during a Mastery-Motivational Climate Motor Skill Intervention: An Exploratory Study"

_ijerph, 2022, doi:10.3390/ijerph192315484_

Round 1

Reviewer 1 Report

Dear editor,

Thank you for giving me the opportunity to review this submitted manuscript.

This is an interesting study. The paper is generally well written and structured. Very few grammatical corrections need to adjust. Also, it was well-founded, with adequate analyzes, and met the aim proposed by authors. However, in my opinion the paper should adress:

Abstract 

1. Please, replace the verb "engage" for "participate". So that it is not continually repeated in the paragraph. (Line 17 or line 20 in page 1).

Introduction 

1. Please, replace "sikllfull learners across" for "skilled learners in" (line 64 in page 2).

2. It is more grammatically correct to say "help participants" than "aid participants". Please, correct the sentence (line 66 in page 2). 

3. Please, replace "with" for "by" (line 72 in page 2). 

4. Where possible, an update of some literature citations would be appreciated. 

Materials and methods 

1. It was well-founded, with adequate analyzes. Also, I would like to stress that the description of the methods used as well as the description of the participants is very clear and concise.

Discussion

1. Please delete the word "directly" as it is repeated (line 289 in page 9). 

2. Please place the connector at the beginning of the sentence "furthemore, we completed..." (line 291 in page 9). 

3. Please, replace "wich" for "that" (line 293 in page 9). 

4. "Engagement" is repeated too many times, please replace for "participation". (line 321 in page 9). 

Conclusions 

1. The conclusion is well drafted. It is also concise and clear. It provides the results of the research and opens up lines of research for future studies. 

References

1. Bibliographic reference number 39 does not appear in the text. Please check the citations. 

2. In several instances I also suggest citing more relevant and recent literature. 

Reviewer 2 Report

First of all, thank you for the opportunity to review this exciting study. We have a rare opportunity to read studies on children in early childhood. As a psychologist, I deeply admire the procedure and effort put in this investigation. The whole body of the text is very well-written. The authors present the gap in the research and knowledge on the structured observation method of child behaviors during motor tasks. The presented aims of the study are clear and innovative. However, I recommend getting rid of the presentation of the accurate results in line 58. I do not have any comments on the Methods, Materials, and Results sections.

In the Discussion, I recommend starting with the description of the results. Then I would present the possible explanations and comparison to the other studies.

Reviewer 3 Report

I commend the authors on an interesting manuscript that provides valuable insights that have not been presented in previous research. The methodological approach is detailed and allows for an in depth analysis of engagement. The writing is clear and concise, with the introduction providing a good overview of the current research and the discussion addresses the knowledge gap on the topic. The intervention program is novel and well described within the text.

While the results graphics provide good detail, they are slightly difficult to read in their current format. I recognise this is not a focus of the current study but it would be interesting to see a follow up on the assessment of skill level of participants after 1, 3 or 6 months post the intervention (i.e., have the children improved?).
